# Community perspectives: An exploration of potential barriers to men's involvement in maternity care in a central Tanzanian community

Nyasiro S. Gibore[1]*, Theodora A. L. Bali[2]

**1** Department of Public Health, School of Medicine, College of Health Sciences, University of Dodoma, Dodoma, Tanzania, **2** Department of Education, Faculty of Humanities and Education, Saint John's University of Tanzania, Dodoma, Tanzania

* nyasiro@hotmail.com, nyasiro2@gmail.com, nyasiro@udom.ac.tz

## Abstract

### Background

Male involvement in maternal health has been linked to positive health outcomes for women and children, as they control household resources and make significant decisions, which influence maternal health. Despite of the important role they have in maternal health care, their actual involvement remains low. The objective of this study was to explore community perspectives on potential barriers to men's involvement in maternity care in central Tanzania.

### Methods

Qualitative research methods were used in data collection. We conducted 32 focus group discussions (16 FGDs with men and 16 FGDs with women) and 34 in-depth interviews with community leaders, village health workers and health care providers. Interview guides were used to guide the focus group discussions and in-depth interviews. The interviews and discussions were audio recorded, transcribed and translated into English and imported into QSR NVivo 9 software for thematic analysis. Three themes emerged from the data; men's maternity care involvement indicators, benefits of men's involvement in maternity health care services and barriers to men's involvement in maternity health care services.

### Results

Both men and women participants acknowledged the importance of men's involvement in maternity health care services, even though few men actually got involved. Identified benefits of men's involvement in maternity health care services include: Learning any risk factors directly from the health care providers and getting prepared in addressing them; and reinforcing adherence to instruction received from the health care provider as family protectors and guardians. Barriers to men's involvement in maternity health care services are systemic; starting from the family, health care and culture-specific gender norms for maternity

**Data Availability Statement:** All relevant data used to reach the conclusions drawn in the study are within the paper and its Supporting Information files.

**Funding:** The authors received no specific funding for this work:

**Competing interests:** The authors have declared that no competing interest exist.

**Abbreviations:** AIDS, Acquired Immune Deficiency Syndrome; ANC, Antenatal Care; AOR, Adjusted Odd Ratio; CL, Confidence Interval; HIV, Human Immune Deficiency Virus; MCH, Maternal and Child Health; PMTCT, Prevention of Mother to Child Transmissions; RCH, Reproductive and Child Health; SDG, Sustainable Development Goal; WHO, World Health Organization.

related behaviour as well as healthcare facilities structural constrains inhibiting implementation of couple-friendly maternity health care services.

## Conclusions

Men's involvement in maternity care is influenced by culture-specific maternity-related gender norms. This situation is compounded by the conditions of deprivation that deny women access to resources with which they could find alternative support during pregnancy. Moreover, structures meant for maternal health care services lack privacy, thus inhibiting male partners' presence in the delivery room. Intervention to increase men's involvement in maternity care should address individual and systemic barriers to men's involvement.

## Introduction

Maternity care is all care in relation to pregnancy, childbirth and the postpartum period. Good maternity care includes: pre-pregnancy and pregnancy care, trained assistance during deliver, knowing the warning signs, planning ahead and community support [1]. All these care require the commitment of both health authorities and community members (husband, male partners, in-laws, parents). Male partners can also contribute to good maternity care by planning ahead and providing support, for example ensuring that women and their families know where to seek help in case of complications, and sharing the workload so that pregnant women can avoid heavy physical effort [1].

Definition of men's involvement varies depending on the context in which it is applied [2]. There is no single commonly used indicator for measuring men's involvement. Different scholars have used varying indicators according to their contexts. According to Bhatta [3], men's involvement is a man attending antenatal care visits, birth plans, encouraging exclusive breast-feeding and immunization for their children. Ditekemena et al. [4] define men's involvement as men's participation in HIV testing during pregnancy. Byamugisha et al. [5] define men's involvement as men attending ANC services with their partners, knowing their partner's antenatal appointment, discussing antenatal interventions with their partners, supporting their partner's antenatal visit financially, taking time to find out what goes on in the antenatal clinic and seeking permission to use a condom during the current pregnancy. Alio et al. [6] consider men's involvement as men being accessible (e.g. present, available), engaged (e.g. care about the pregnancy and the coming child, want to learn more about the process), responsible (e.g., is a caregiver, provider, protector), and maintaining a relationship with the woman carrying the child regardless of their own partnership status. In this study, men's involvement is defined as men attending antenatal care (ANC) visits and relieving their pregnant partners from performing heavy workload.

For a number of decades, advocates for improving maternal and child health have promoted men's involvement in all aspects of health care for women and children [7]. The effect of men's involvement on health outcomes for women and children is associated with their knowledge, education, attitudes and behaviours [2, 8–10]. Lack of knowledge regarding complications and danger signs in pregnancy and childbirth has been frustrating for male partners and this has created barrier to their involvement in maternity care issues [11]. Lack of knowledge notwithstanding, men as decision makers, partners and parents, have important role in the preparations for birth and influence actions needed in case of an emergency within their households and communities [12]. In most African countries, especially those in patriarchal

system, men control household resources and make significant decisions that influence maternal health and the choice of when and where a woman should have access to health services [13, 14]. In such situations where there is disregard of women's views and deprivation of women's access to household resources, women's and children's health status is affected.

Globally, men's involvement in maternal health has been linked to positive benefits, such as increased likelihood of contraception use [15]; increased uptake of interventions to prevent HIV transmission [16]; increased access to antenatal and postnatal services [17]; enhanced gender equality; and empowerment of individual women to make their own decisions [18]. Men as individuals can also benefit from being involved in parenting and reproductive health programs [19], support good nutrition, reduce workload during pregnancy, assist in birth preparedness and provide emotional support [20, 21]. Several studies have shown that interventions to include men in maternal health issues improved maternal and child health outcomes. For instance, Yargawa and Leonardi-Bee in their systematic review and meta-analysis of men's involvement found that, men's involvement increases skilled birth attendance and use of postnatal care and also it was significantly associated with reduced odds of postpartum depression among women [22]. A randomized control trial among couples in Burkina Faso found that, couples who participated in the education-focused intervention were more likely to be exclusively breast-feeding three months postpartum and also use effective contraception eight months postpartum [23]. The study conducted in Southeast Nigeria found that men who were aware of female contraception had three times greater odds of having spouses who wanted to use contraception compared to men who were not aware of female contraception [24]. In Tanzania, the intervention to educate pregnant women's husbands on danger signs and pregnancy complications resulted in a significant increase in birth attendance by skilled health providers —i.e. 51% for the intervention group compared with 34% for the control group [25]. While several studies have reported the benefits of including men in maternity care, the arguments on the negative side of men's involvement have also been made including discrimination against women, increased male dominance in decision-making and marginalization of married women [2, 26]. Others include the potential for escalating labor difficulty when husbands become anxious in delivery rooms [17], decreased utilization of PMTCT services among unmarried women when required to come with their partners [27] and failure to return for the second visit when asked to come back for ANC services together with their male partners [28]. In situations where men are against contraceptive use, women may face difficulty accessing contraception.

In spite of the important role that men play in maternal health care, their actual involvement remains low in most African countries [29, 30]. Lack of men's involvement in maternal health care may contribute to delays in progression towards the achievement of Sustainable Development Goal (SGD) of reducing maternal mortality. Several studies in Africa have explored men's involvement in maternal health programs such as prevention of Mother-To-Child Transmission (PMTCT) [5, 16, 31, 32]. However, studies investigating men's involvement in maternity care and their barriers in Tanzania are limited. A study conducted in Uganda to explore perceived benefits of men's involvement in antenatal care found that knowledge of three or more antenatal care services, obtaining health information from health care workers and a spouse having attended a skilled birth at last childbirth predicted increased men's attendance at ANC [33]. One study conducted in Malawi found that using peers to encourage men's involvement was effective and sustainable in increasing male involvement in maternal care [34]. In Kenya, a study to explore the barriers and opportunity for men's involvement found that men were aware of the benefits of their involvement in maternal health. However, perception of pregnancy support as a female role, negative health worker attitudes toward men's involvement, and unfriendly antenatal care services limited men's

involvement [35]. Some scholars have reported that gendered norms and expectations hinder men's involvement in reproductive healthcare and their access to programs and services [36, 37]

In Tanzania, men's involvement in maternal health care is a new idea [38]. Pregnancy and childbirth issues have traditionally been regarded as women's affairs and pregnant women were given support during labour and childbirth by either their mother in laws, sisters or other women relatives [39]. Men were mainly responsible for providing money for medical bills and other material needs as well as naming the newborn [31]. Although maternal and child health policies recognise the importance of engaging men in reproductive and child health [40], studies have reported low male involvement in maternal health care [41–43]. Among the factors reported to hinder men's involvement in pregnancy and childbirth were traditional gender roles at home, fear of HIV testing and unfavourable environment in health facilities [38]. However, less is known regarding the in-depth understanding of community perspectives on barriers to men's involvement in pregnancy, delivery and postpartum care in Tanzania. The few available studies have explored the perception of men's involvement in pregnancy and childbirth only and most of them were health care facility based studies. To address this gap, this study explored community perspectives on potential barriers to men's involvement in maternity care in Dodoma Region, central Tanzania. We explored how gender normative assumptions such as pregnancy and childbirth being women's responsibility, have shaped policies and how these assumptions have been deconstructed by the emergence of the men's involvement agenda. The study contributes to literature that informs the integration of normative gender behaviour surrounding maternity and awareness of structural blind spots in the maternal health care policy.

## Materials and methods

### Study area

The study was carried out in Dodoma Region of Tanzania which is located in the central part of the country. It involved four out of seven districts in the region, namely Kongwa, Kondoa, Chamwino, and Dodoma Municipality. The districts were selected based on diverse characteristics of each in relation to men's involvement. According to the documented Regional Reproductive and Child Health Coordinator's annual report of 2011 (unpublished), among the seven districts in the region, Kongwa District was doing the best to motivate men to accompany their partner's to the first ANC visit. It had 99.9% of men's attendance with their partners. Kondoa District had 24% of men's attendance with their partners. Dodoma Municipality had 13%, while Chamwino District had not reported any information related to men's involvement in ANC. Therefore, the sites were selected purposeful in order to understand the perspectives of those men who were highly motivated to provide companionship and those who were not attending with their partners.

### Study design and population

A cross-sectional study design with a qualitative approach was employed as little was known about the barriers to men's involvement in maternity care and information was gathered one time, across community segments. Also, there was a need to understand the context of men's involvement in maternity care. The study population consisted of living together married couples, health care providers, village health workers and community leaders.

## Sampling procedure

A multistage sampling technique was used for selecting the sample units. The four districts were purposively selected. Within each district, two wards (sub districts) were randomly selected from each district using a table of random numbers to make a total of eight wards: Mnenia and Pahi (Kondoa), Mlanga and Makang'wa (Kongwa), Mlowa Barabarani and Chinangali-II (Chamwino) and Msalato (street) and Nzuguni (street) (Dodoma Municipality). Finally, one village or street (in case of municipality) was selected randomly from each of the eight wards using a table of random numbers, maintaining the total of six villages and two streets. From the villages and streets, 32 households (four households from each village/street) were identified based on the following criteria; married couples who had children aged two years or below at the child's last birthday. The men must have resided with their spouses in the same household and their partners must have had a second or more pregnancy at the time of data collection. Couples who met the inclusion criteria but refused to participate and those whose partners were not mentally fit to be interviewed were excluded in the study.

The households were purposefully selected and couples were recruited to participate in the study by the principal investigator with the assistance of hamlet and street leaders. Principal investigator explained the objective of the study to participants before asking for their participation. Participants were invited to participate in the focus group discussion which were conducted separately for male and female. Health care providers (in-charges of the health facilities, heads of reproductive health services), village health workers and community leaders residing in the study area, who were available at the time of data collection, were included in the study.

## Ethical considerations

The approval to conduct this study was granted by The University of Dodoma Research and Ethical Clearance Committee. Permission to collect data was obtained from Dodoma Regional Commissioner, Dodoma Municipal Director, District Executive Directors of Kondoa, Kongwa, and Chamwino district councils as well as District Medical Officers of all four districts and the local leaders of all wards. A free informed consent (written or verbal) was obtained from each individual participant at the start of the study.

## Data collection

Data were collected through FGDs and IDIs, using the interview guides. Prior to data collection process, the interview protocols were pretested and adjustments were made according to early experience and information participants had provided. The interview protocol consisted of questions and prompts to guide the sessions. The village that was involved in the pretest was not included in the actual study. Participants in the focus group discussions were approached face to face through the guidance of hamlet and street leaders. Health care providers, village health workers and community leaders were contacted through letters and telephone calls. A total of 32 FGDs (16 for men and 16 for women) comprising of seven to nine respondents were conducted. A total of 246 participants (123 couples) were contacted for focus group discussions, out of which 236 participants (118 couples) agreed to participate in the discussions. Men were involved in men's focus group discussions and women in female's focus group discussions. Five couples (10 participants) declined to participate due to their busy schedule. Twenty four (24) community leaders, 15 health care providers and five (5) village health workers were contacted for in-depth interviews. On reaching saturation after 20 interviews with community leaders, 11 health care providers and three (3) village health workers, no further

attempt to interview the remaining participants was necessary. Therefore, a total of 34 in-depth interviews were conducted. Two community leaders declined for lack of time.

The first author moderated all the FGDs, the second author acted as an observer for the FGDs while three trained research assistants (one female and two males), all married, took notes and operated the tape recorder. The male research assistants were involved in men's focus group discussions while the female research assistant was involved in women's focus group discussions. The second author conducted IDIs and first author took notes and recorded the interviews. The FGDs were conducted in Kiswahili.

Research assistants were chosen based on their experience in interviewing and fluency in Kiswahili. All were trained for three days to interpret and deliver the interview guides in the same manner in order to draw out consistent information. After self-introduction, the objective of the study was explained to participants and they were allowed to ask any question related to the study objective. The session started by asking participants how their spouses and children were, then went directly to asking questions related to the study objective. After every FGD and IDI session was completed, a member checking technique was used to ensure that what was heard or written was correct and the transcripts were returned to participants for comments. This technique involved restating what was recorded, summarizing or paraphrasing the information from FGDs and IDIs.

The FGDs took place in the village or ward offices or in the nearby classroom, depending on the convenience of participants. The IDIs took place in the hamlet or street leader's offices, the health care provider's offices and village health care worker's offices. No any other person was allowed to the discussion apart from the researchers and participants. Saturation point was reached when no new information related to the research topic was emerging. Thus after our discussion as researchers we stopped collecting more information. Participants' age ranged from 18 to 70 years. Most participants (70%) were self employed and mainly engaged in farming activities. About 77% had completed primary level of education.

FGDs provided a chance for participants to discuss their views regarding responsibilities of men during maternity period, their perspectives regarding maternal care during maternity period and cultural practices related to pregnancy and childbirth in the community. Other advantages include discussion on their participation in maternity health care services, benefits of including men in maternity health care services, and barriers to men's involvement in maternity health care services. Interviews lasted between 30 and 45 minutes, while focus group discussions lasted between 50 and 70 minutes. The data were collected between December 2016 and June 2017.

## Data analysis

Interview data were transcribed verbatim and translated into English language by two research assistants who were fluent in both Kiswahili and English. Back translation was done to keep the original meaning of the data. The second author proofread the transcripts to check for accuracy. Data were analyzed thematically. The first author (NSG) imported transcripts and field notes into QSR NVivo 9 software to facilitate the coding process. After all necessary files were transferred, the coding was done, the process of putting together extracts that are related to each other into basins called nodes. The analysis was carried out in three steps. First step, thorough reading of the transferred transcripts into QSR NVivo 9 software and creating nodes in the process to house relevant excerpts or text from the transcripts. The emerged major issues were noted and coded according to the meaning they conveyed. Additional codes which emerged during coding were added after consensus of both authors and saturation was attained when no more codes emerged from the transcripts. In the second step, authors

assembled the codes into different themes and a thematic chart was developed. Patterns, similarities and differences in these codes and themes were examined to ensure that all views were considered in the research report. Codes that expressed similar meanings were grouped together and assigned a phrase that depicted the common message. The phrases from several groups of codes constituted description of themes. The second author checked the accuracy of themes descriptions to find out if they had not been under or over represented. The third step involved contextualizing the original codes along with the text so as to answer the aim of the study. This process resulted in more abstract analytical themes. Descriptive text (tale) was applied around the analytical themes by quotes used to illustrate the text in order to converse its meaning to the reader.

Data analysis was performed by each author separately. They then compared their outputs, and reached an agreement on the codes and themes. The results generated by FGDs and IDIs techniques were triangulated. Three main themes that were finally identified were: men's maternity care involvement indicators such as men's attendance at maternity health care services and workload support to women during maternity period; benefits of women having their partners at maternity health care services; and barriers to men's involvement in maternity health care services.

## Trustworthiness of the study

To ensure quality of the data the criteria of credibility, transferability, dependability, and confirmability were adhered to. Credibility was ensured by triangulation of data collection methods whereby both FGDs and IDIs were used, a voice recorder was used to capture all interview sessions, and field notes were taken to supplement what was not recorded. Prolonged engagement in the study field, whereby the researcher (first author) stayed for more than six months prior to data collection to become familiar with the settings. She was not known to the participants of this research prior to undertaking the study and she did not undertake any clinical or teaching activities locally alongside this research. Whilst it was useful to understand what the community was talking about men's involvement in maternity care, as a researcher she made attentive efforts not to accept possible common assumptions at face value. This increased the credibility of the data source. Thick descriptions of the research methodology that contain important information a reader may need to know in order to understand the findings, the use of purposeful sampling which enabled the selection of information-rich participants who fulfilled the criteria for participation and encouragement of confidentiality throughout the study period, all this ensured transferability of the information collected. Dependability was ensured by using an emerging design analysis where, new issues that emerged were considered in subsequent data collection and in the analysis process. The involvement of more than one researcher in the study process ensured that the interpretations emerged in data through researchers' triangulation, which also enhanced dependability of the data source. Furthermore, both authors are Kiswahili native speakers. The first author is expert in public health while the second author is expert in applied anthropology and community development work. They were both involved in data collection, which also provided dependable data. A member checking technique was employed during data collection to meet the criteria of confirmability where the researcher restated, summarized or paraphrased the information received from IDIs and FGDs to ensure that what was heard or written down from the interviews and discussion was accurate. The translation and back translation of research tools from English to Kiswahili and then Kiswahili to English was intended to increase the free expression of participants in Kiswahili and to check accuracy of translation respectively. This enhanced confirmability of the data.

## Results

After the data analysis, three main themes were generated namely, men's maternity care involvement indicators such as men's attendance at maternity health care services and workload support to women during maternity period; benefits of men's involvement in maternity health care services and barriers to men's involvement in maternity health care services as demonstrated in Fig 1. Six barriers to men's involvement in maternity health care services were identified, including: culture-specific gendered expectation for maternity-related behaviour, pregnancy outside wedlock, fear of HIV testing, precarious family and economic situations, lack of knowledge and inadequate information, and health care system-based factors.

### Men's maternity care involvement indicators

Perspectives of men and women participants seem converging on why men should get involved in maternity care. They generally found it important for ensuring quality maternity care for women and their newborn babies. Indicators of men's involvement in maternity care included men's attendance at maternity health care services and men's household workload support during maternity period.

### Men's attendance at maternity health care services

During in-depth interviews and focus group discussions, it was noted that among men who escorted their partners to maternity health care services, most of them reportedly attended during the first visit of antenatal care. During the second, third and fourth visits very few men attended with their partners. During delivery most women were accompanied by their fellow women. A female healthcare provider from Kondoa explained:

> "*Only few men do accompany their wives. . ., and most of them turn up only during the first day. Others who are also few come during the second, third and fourth clinic visits. As the number of clinic visit increases, the number of men who attend decreases. During delivery period, only one out of hundred men accompanies their wives. Most of them are accompanied by their female relatives.*" (IDI-1, female health care provider)

Some male participants opined that men accompanied their pregnant partners during the first antenatal visit because either they were eager to know the HIV status of their partners or it was mandatory that they should check their health status (HIV) with their partners. A male participant from Chamwino said:

> "*Men accompany their wives, if a woman is pregnant. This is like a rule, because all of them are required to check their health as early as possible. But, if it is a matter of accompanying her while attending clinic or taking a child to hospital; they will never do so. But when the pregnancy is at its early stages and you are required to report to clinic, in such a situation we go together.*" (FGD-24, male participant)

Men were willing to accompany their pregnant partners in the first antenatal visit as part of being obligated. This was usually not the case with other subsequent visits where they did not show up unless there was a serious issue they needed to settle. A young lady from Dodoma Municipality explained:

> "*I can take my husband to clinic, or I can go alone, and he can agree. He can accompany me only one day. He will never show up the next visit unless he is required for medical check-up*

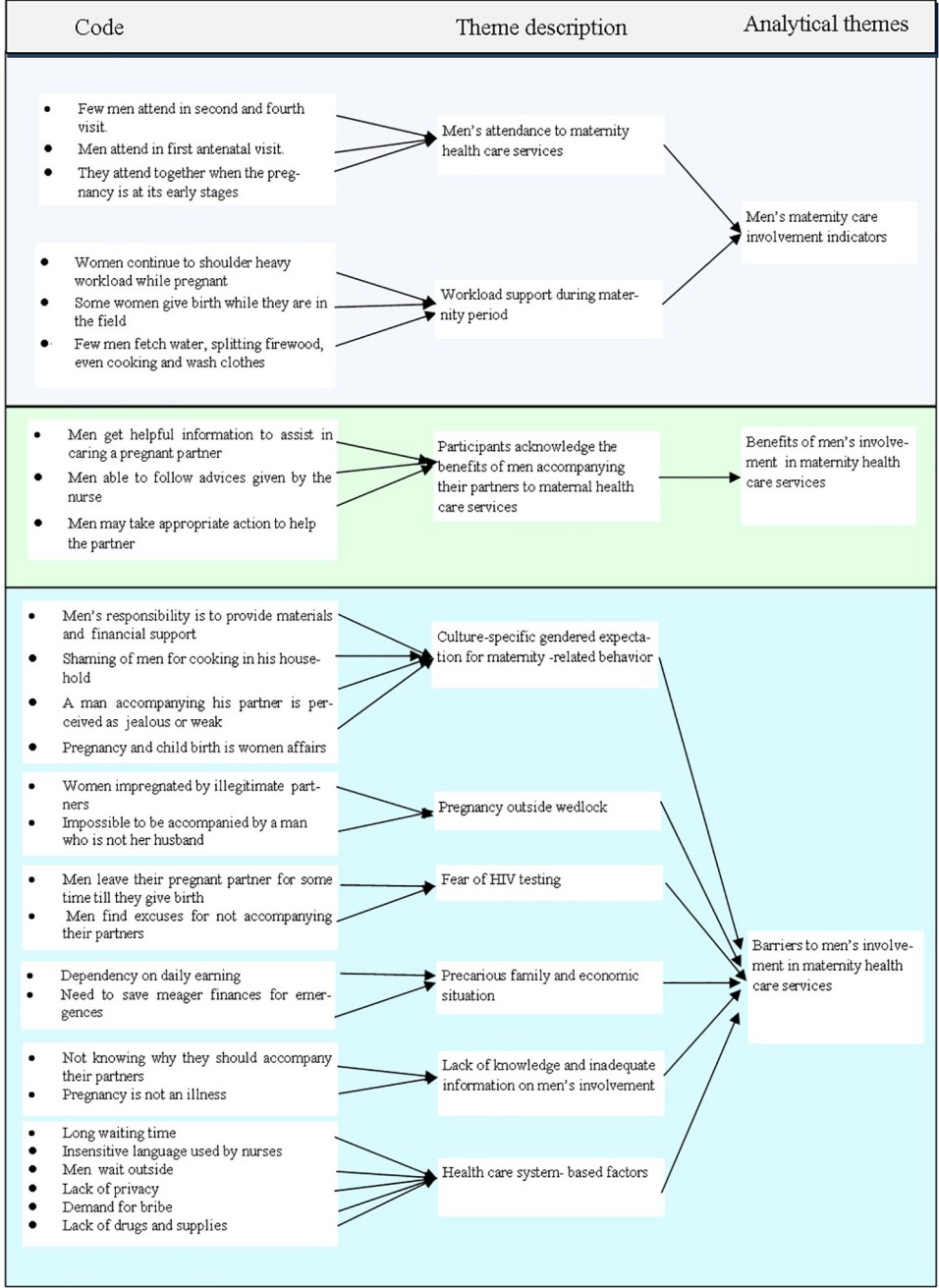

**Fig 1. Example of thematic analysis from codes to analytical themes.**

*which involves both parties or if there is an issue he is needed to contribute. Otherwise, he never agrees to accompany me for every clinic visit." (FGD-31, female participant)*

Another man from Kondoa explained:

*"For us here in Pahi I would like to say that if they (partner accompanying men) are there, they are very few. It does not happen often to accompany our wives to maternal and child health services. Only few men do so." (FGD-5, male participant)*

## Men's household workload support during maternity period

During the FGDs both women and men reported that only few men actually reduced the workload of their partners during maternity period. Most women reported having worked the same usual or even more than usual household workload. They also reported working in agricultural activities up to delivery. For instance women explained:

"*A woman works from the beginning to the end of her pregnancy.*" *(FGD-14, female participant)*

"*Those who assist their wives are few. You may find that during the cultivation period a pregnant woman continues to cultivate in her farm and it reaches a time when some of them deliver at the fields. You can continue working up to delivery and after delivery you stay for a week and then you carry a bucket of water on your head, firewood and the baby at your back. If it is during cultivation season, you stay for three days and the fourth day you go to the farm with your newborn baby.*" *(FGD-24, female participant)*

It was noted that women got assistance mostly from a female relative rather than their male partners. For instance, in one focus group discussion a man narrated:

"*The local families within these areas live like relatives; you will therefore find them assisting each other. But most of them are women who are very close. A woman is in a better position to know what her colleague suffers. Very few men can afford to do so.*" *(FGD-2, male participant)*

Similarly, a woman from Dodoma Municipality explained:

"*In our family, once a woman is pregnant her partner will call his mother, or sister to assist the woman. But it is difficult for her husband to take part like a woman. It becomes impossible.*" *(FGD-32, female participant)*

During the discussion with men, very few reported to have shared more work than usual compared to the times when their partners were not pregnant. Majority of men reported to have shared the same or less amount of work. For instance, in one focus group discussion a man said:

"*The issue of relieving a pregnant woman from work, only few of us do. In fact others will tell you that, they do not see any reason for relieving her; they can say let her work. She can even fetch water and carry tedious works, but the husband does not care.*" *(FGD-17, male participant)*

In women's focus group discussion, a woman said:

"*No assistance so far. You will know yourself how to accomplish your duties until you give birth; then they will tell you that is part of physical exercise.*" *(FGD-7, female participant)*

## Benefits of men's involvement in maternity health care services

Both men and women participants seemed to realize the benefits of men's involvement in maternal health care services. They agreed that it was important for a male partner to provide

a companionship so that he can hear the first hand information from the health care provider which will help him to take appropriate action when need arises. Women in the FGD expressed their desire to have their partners with them in the clinic at the time when the health care worker discusses pregnancy problems during clinic attendance. For instance one female respondent narrated:

"*When we attend at the clinic together he will be told by the health care worker the possible problems which can arise during my pregnancy and he can take appropriate action to help me when complications arise. Also he will be able to follow all the advices given by the nurse because he was told directly by the nurse, instead of remaining at home and waiting for me to explain to him what happened at the clinic. Some men are reluctant when told by their partners what happened at the clinic.*" *(FGD-3, female participant)*

Men also view their participation in maternity health care services as important because they would be furnished with information that would help them to take care of their pregnant partners and take appropriate action when problems arises. Also they would be able to remind their partners to adhere to what they had been instructed by the health care provider. A male participant from one focus group discussion narrated:

"*If I accompany my wife to the clinic, and she is advised that she should eat fruits or use this or that medication, when she reaches home, she might forget, but if both of us are listening to the doctor, we will be reminding each other what we were told at the clinic, and my job will be to make a follow up on what we were advised by the experts*".*(FGD-25, male participant)*

Another male participant added

"*First, I will know the problems of my wife; and secondly, I will know the progress of my expected child. As a man if I am told there is any deficiency, it is my responsibility to be prepared because I have heard and seen what I was told at the Health Centre*".*(FGD-9, male participant)*

## Barriers to men's involvement in maternity health care services

The involvement of men in maternity health care services was influenced by factors related to culture-specific gendered expectation for maternity-related behaviour, pregnancy outside wedlock, fear of HIV testing, precarious family and economic situations, lack of knowledge and inadequate information, and the health care system-based factors.

## Culture-specific gendered expectation for maternity-related behaviour

During the FGDs and IDI it was noted that, although men's involvement in maternal health care was acknowledged by both men and women, cultural conceptualization associated with pregnancy and childbirth in the society is complicated. Respondents in most focus groups expressed the view that when it comes to the issue of companionship to maternal health care services, they were generally involved in arranging transport for women to attend the clinic and provide financial and material resources. Also, they were directly involved in critical situations such as when the child or woman was seriously sick. But, in ordinary situations, pregnant women were accompanied to maternal health care services by their fellow women. For instance, one of the male community leaders said:

"*According to our tradition and customs, usually a mother takes care of the children and not the father. A woman takes care of her pregnancy as well as taking children to hospital. The husband will only give support if his wife or a child is in critical condition. That is why when a man takes a child to hospital people will be asking themselves*: "*Doesn't this man have a wife*?*" (IDI-29, male leader)*

Some respondents were of the opinion that according to their culture it is shameful for a man to walk along with his wife in the street. If a man is seen escorting his partner to the clinic or anywhere, the community will judge him as a weak man and as being controlled by his partner.

"*This is due to the practice that existed and inscribed in their minds from a long time, . . . This means according to tradition and customs or patriarchy, if a man goes to clinic, he will be seen by the society that he is controlled by his wife or he is submissive to his wife. At this juncture, a man could fail to give positive support to his wife for fearing that the society will say he is controlled. . . ." (IDI-9, male leader)*

From Dodoma Municipality one man said:

"*In addition, it is only customs that prohibit us. As my father there said, going to the clinic together with your wife, while people see you carrying a baby, leaving your wife stroll alone, according to us, we can say you are kept under the control of your wife." (FGD-26, male participant)*

As a barrier during FGDs it was also reported that men who escorted their wives were stereotyped and subjected to gossip by their male counterparts and interestingly also by women at the clinics. This resulted to negative views like, a husband being too swamped in love or jealousy, a husband being controlled by the wife, the clinic being a domain of women; not men, and that, women only should escort fellow women to clinics. For instance, a man from Chamwino said:

"*When you attend the clinic with your wife, others will say you are jealous of her. Others will say she gave you a love portion (limbwata which means local medication) that makes you look a fool. They will say, "Look he carries a baby on his back and he is going along with his wife to the clinic." (FGD-18, male participant)*

Another male respondent added:

"*Others will say, the husband is an idiot, the wife overpowered him (locally known as "bushoke"). But they do not know that is the kind of marriage life the couples are supposed to live." (FGD-17, male participant)*

When it comes to the issue of workload support to pregnant women, participants in several focus groups discussions were in consensus that most women continued to shoulder heavy workload during pregnancy. The reluctance of male partners to alleviate their pregnant partners of heavy workload as well as household chores was attributed to traditional male-female social expectation for gender roles. One of the male participants from Gogo tribe stated as following:

"*It is abnormal. Very shameful! I have never cooked in my life. It is really a shame if someone finds me cooking in the house. It is much better if I look for my sister or my daughter or any housemaid who will cook. Honestly, I have never done it and I see that it is shameful!*" *(FGD-26, male participant)*

Another man added:

"*When it comes to cooking, I disagree; we do not assist each other. In our custom as Gogo tribe most of us do say*: *'I cook!! Am I a woman'? We do not assist them in cooking.*" *(FGD-17, male participant)*

This may indicate that women are not enjoying privileges in the household during maternity period due to lack of workload support from their male partners. Little or lack of support during maternity period may signify that a woman need to work extremely hard with incidents of premature labor and all other birth related complications. For instance in one focus group discussion a woman narrated:

"*There are those who give birth in the field, because labor pain comes suddenly, you never know. You come from the farm with your husband's hoe and yours along with your child on your back while you are also pregnant. The husband walks here and there. He does not care. At times labor pain may start while you are in the field and you will find yourself delivering at the field.*" *(FGD-23, female participant)*

The evidence like these may show some of the difficulties pregnant women face regarding workload support and the lack of flexibility in task negotiation within the household. This could happen for the reason that men have no full understanding of women's pregnancy affairs or because they are afraid of been mocked at when they participate in household chores. During the interview with community leaders, it was noted that there are few men who were able to relieve their partners of workload. However, this usually happened after their partners have delivered. A male community leader added:

"*During pregnancy most women keep on working. It is only when they deliver, they receive a little support.*" *(IDI-2, male leader)*

This could mean that after delivery men are happy of the new life brought by women in their household. This may trigger them to take more responsibilities so that a woman can have some time for caring their new baby. When asked about their specific domestic tasks during their partner's pregnancy, men reported washing dishes and clothes. But, cleaning the house and cooking were not their favorites. Childcare, fetching water, collecting fire wood and charcoal were more likely to be carried out, although this contribution is still relatively low. For instance, three of the male respondents had this to say:

"*Few of us fetch water, split firewood, even cook and wash clothes.*" *(FGD-5, male participant)*

"*I may take care of other small children.*" *(FGD-29, male participant)*

"*Yeaah... When my wife is pregnant, once she reaches her seventh month and onward, I always take responsibility to perform all domestic activities, such as washing clothes and attending our children to take bath. Although I am instructed by doctors that she should do*

*physical exercise, but at home, I used to assist my wife several times, in particular, those tedious and important activities." (FGD-6, male participant)*

This kind of evidence and other similar stories can explain how pregnancy and childcare are viewed exclusively as women's responsibility. In this community, there is strict gender role distribution when it comes to the issue of taking care of deliveries and childbirth. Female gender is responsible for maternity-related functions including accompanying a pregnant woman for routine ANC attendance and delivery as well as taking care of a delivered mother and her newborn baby. The type of care includes looking after the newly born baby, cooking and washing for the mother and helping her with all household chores. The male gender's responsibilities are providing for and protecting the family including arranging transport and covering all issues that will facilitate a woman to deliver in the health care facility; purchasing the items that are needed for a woman to deliver; and saving money for emergencies.

## Pregnancy outside wedlock

During focus group discussions and interviews with study participants, it was noted that pregnant women were required to attend the first antenatal visit with their partners. There was an allegation that, those who attended without their partners were denied services. However, pregnant women with genuine reasons for not being accompanied by their partners were required to present a letter of exemption from their village leaders. For this reasons, when it comes that a woman got pregnancy outside wedlock, it was difficult for a man to accompany his pregnant partner to the health care facility. Therefore, this created barriers to men's involvement as well as for pregnant women to start early booking of antenatal clinic. This was emphasized by one of the female health care provider during the in-depth interview:

"*A woman could have been impregnated by somebody's husband, in such a situation she would never come with him here." (IDI-24, female health care provider)*

A similar view was reported by nearly all focus groups of men and women. One female respondent had this to say:

"*There are some cases; a significant number of women will say, 'He is not my husband. How can I ask for his company to the clinic'?" (FGD-28, female participant)*

This finding is interesting given that the study sample consists of married couples with a child less than two years of age, living together at the time of fieldwork. It is a more intriguing finding given that most other responses seem to suggest that male partners are generally in control of most decisions including those related to reproductive matters.

## Fear of HIV testing

Information elicited from the FGDs also showed that men felt being forced to attend reproductive and child health care with their partners for prevention of mother to child transmissions (PMTCT) couple's counseling and testing for HIV. Majority of the respondents opined that, men were not attending the clinics with their partners because they are afraid of undergoing HIV testing. Fear of stigma in case they test HIV positive made some men to leave their households and come back when their partners have delivered. For instance, a female respondent from Chamwino said:

"*May be others are afraid if their blood could be tested for HIV. They used to say that, 'Once we go to clinic together and the results show that I am HIV positive, I will be shocked. Therefore, I am not going to attend'. When a woman's pregnancy reaches its first or second month, the husband leaves her and goes somewhere and comes back when he hears that she has delivered.*" (FGD-20, female participant)

Similarly, during FGDs with men the fear of HIV status disclosure was further reiterated as one man commented:

"*Another issue which is very common in many villages is the notion that, once the husband is asked to accompany his wife for medical check-up the HIV test would not be an exception for him; those are the feelings. But he does not know that when you attend to the clinic with your wife, you have an opportunity for examining maternal and child health as well as yours. This is also among major problems! I agree with what this old man said, lack of education affects most villagers.*" (FGD-25, male participant)

## Precarious family and economic situations

During the FGDs it was noted that some men realized it was their responsibility to accompany their partners to reproductive and child health clinics and give physical and social support when needed. But this was prevented by men's job responsibility because men were the ones who are the providers in the family. Men further articulated that their absence at work and being present in the maternal health care services together with their partners could leave children without food, as many felt that one day or few hours' absence from work would mean a struggle for survival for the next day. Men also were concerned of the responsibility of taking care of other children when their partners are absent. For instance, one of the male respondents narrated:

"*Bus fare is the problem, but it is not the biggest as such. The major problem is the management of time and the kind of work we are doing because we have to work every day. In so doing, probably I can get five thousands shillings a day. Therefore, if I do not work today, I will miss such sum of money.*" (FGD-26, male participant)

Another male respondent from Kondoa added:

"*Yes!! The economic situation of a person is an added factor. Not all people are economically strong. . .only few of them are in that position. Others are economically poor. Therefore, they cannot afford to go together to the clinic. Sometimes a man can let his partner go alone for fear that, if they both visit the clinic they will face financial problems.*" (FGD-5, male participant)

## Lack of knowledge and inadequate information

During the in-depth interview with health care providers it was reported that there was a health policy in place to motivate men to accompany their pregnant partners for antenatal care. This was mainly due to the need to reduce risk of infecting the fetus through requiring both partners to be tested for HIV. Apart from testing for HIV at the antenatal care, men who attend there have an opportunity to access critical information related to maternal health such

as obstetric danger signs and birth preparedness. As head and decision maker in the household, information obtained at the antenatal care may result in increased observance of guidance offered at the clinic. Most men reported that they were unaware if they were called to attend at the ANC with their pregnant partners and that they did not understand why they should go to the clinics with their partners while previously women used to go alone. Even for those who were aware of the need for their attendance, few accompanied their partners. This was emphasized by some of the respondents:

> "*Let me explain why people do not attend clinic. In the past people were not informed on the importance of husband and wife attending together. That is why there is such weakness.*" (FGD-6, male participant)

Another male participant added:

> "*That is how it is, we do not know why we have to attend clinic, and what kind of advantage we can get*" (FGD-10, male participant)

Both men and women participants in several focus group discussions were of the view that lack of knowledge on maternal health matters and inadequate information on what is being done at antenatal clinic keeps men away from taking their responsibilities as partners in reproductive health. They argued that men were not knowledgeable on what they were supposed to do when their partners become pregnant and they did not have appropriate information to guide them. One of the female respondents emphasized:

> "*Failure to get education and information makes others to feel that they have no responsibility to their wives. But those who are educated are aware that, once his wife becomes pregnant, he has the responsibility to take her to the clinic for medical check-up in order to find out if she has any problem. If the result shows that she has problems he has the duty to ensure that she gets proper treatment. However, there are some husbands who think that attending clinics is the wastage of time. He may also find that he has no time to know advantages and disadvantages of going to clinic with a partner.*" (FGD-19, female participant)

One of the community leaders added:

> "*Men are not informed if they are supposed to accompany their wives to clinics and they are not even aware of what is being done at the clinic.*" (IDI-17, male leader)

### Health care system-related factors

A number of health care system-related factors influencing men's involvement in maternity care were identified in this study. These include long waiting time for the services at the health care facilities, distance to the health care facilities, insensitive language used by health care providers, tiny consultation rooms and little privacy in labour wards, shortage of medical supplies and demand for bribes.

### Long waiting time

During the FGDs, it was noted that men's limited time to be in the clinics together with their partners, coupled with concurrent job demand was further complicated by long waiting time for the services. Many female respondents expressed concerns that they spent long time in

clinics to receive services and their partners cannot just waste their valuable time in clinics for hours waiting for the services. This was emphasized in some of the FGDs responses:

"*Some may attend at the clinic. However, as they find that it takes much time to get services, they lose hope and get back home or go to their other activities.*" *(FGD-5, male participant)*

A female health care provider added:

"*Failure to get services timely when they attend clinics discourages them (men) to come back again.*" *(IDI-13, female health care worker)*

Some of the respondents expressed opinions that shortage of health care providers was the contributing factors for long waiting time in receiving the services at the health care units:

"*Shortage of doctors and nurses at clinics is among the major factors resulting to delays in getting services. For example, a person would sacrifice the whole day for clinic, say, from morning to evening; not to mention that they leave other children at home. Who would take care of them? This is also a serious problem contributing to poor attendance of men and some women to ANC in this area.*" *(FGD-6, male participant)*

"*Shortage of health care providers where many people attend at the clinic is another factor. Sometimes, it happens that people are served by only one attendant; as a result some people lose patience when they find that they have stayed at the clinic for more than four hours that's to say from ten in the morning to two in the afternoon. Being attended by only one employee is a problem!*" *(FGD-5, male participant)*

## Distance to the health care facilities

Distance to the health care facility and lack of reliable transport were reported to create barrier to men's involvement in maternity health care services. It was noted that walking and occasionally riding or cycling were the main means of transport to reach most of the maternal health services. Men reported being unable to access the services due to its distance and that the time spent to travel to and from the health care facility, accompanying their pregnant partners who are not sick could be used for farming activities and other responsibilities. One male participant said:

"*Health care units for maternal and child health services are far; therefore, most people fail to access them.*" *(FGD-5, male participant)*

It was also reported that in situations where pregnant women were not able to walk a long distance due to illness or labor pain, it was when they get companionship from their male partners by being carried on the bicycle, motor cycle or any other kind of transport. One male participant shared his experience:

"*Our clinic here is very far; she is the one who is pregnant and need to be examined. I cannot accompany her for every clinic visit. I have other farming duties to do. But if she is near to giving birth and not able to walk alone, I always hire a bicycle to ferry her to the hospital*"*(FGD-1, male participant)*

This narrative illustrates how accessibility to the health care facility limits men's ability to access health services with their female partners, forcing female partners to attend alone simply

because they are the ones who are pregnant; thus directly facing the impact of not attending services at the health care facility.

## Insensitive language used by health care providers

FGDs respondents pointed out that health care provider's attitude and behaviour toward mothers and their partners impede men's involvement in maternity care. Respondents reported that some of the health care providers were rude, used bad language and had no respect to clients. For instance, one woman narrated:

> "*There are some areas here our nurses are cruel and arrogant. When you reach the clinic, they will tell your husband and criticize him for impregnating you frequently and make you deliver without supplying basic needs such as clothes, food, etc. These words have negative effects on men's impression of maternity health care clinics. Husbands become annoyed, and they tell their wives to choose clinics of their own. But if the nurses could use polite language, our husbands will support us and they would not let us come alone.*" (FGD-16, female participant)

A female community leader from Kongwa supported this line of argument saying:

> "*Harsh statements from nurses discourage men. For example, when a woman becomes pregnant at short intervals, nurses attack men verbally and harshly. As a result they are afraid to come back again.*" (IDI-16, female leader)

Although the policy for ANC prioritizes services for couples, the testimonies like these above may make men to feel that they are not part of ANC attendees as they are neither acknowledged nor received well.

## Tiny consultation rooms and little privacy in labour wards

Participants pointed out that one among health care facility barriers was difficulties of being in clinics together with their partners due to limited space to accommodate both parties in the consultation room. They argued that available space in the consultation rooms generally accommodated two people only (meaning the health care provider and one client); there was no space for the third person. It was reported that, often than not there was only one chair in front of the health care provider's desk. It was also reported that most labor wards were open and had not constructed in cubicles. If men were allowed in labor wards to be with their laboring partners, they would possibly see other laboring women. This was mostly regarded as unethical. It was lamented that, few men who escorted their partners to clinics were at times asked to stay out of the labor ward and they hung outside waiting for outcome of their laboring partners. For instance, one female health care provider explained:

> "*The environment for clinical services is not conducive. The consultation rooms are small. . ., cannot accommodate both wife and husband. And the maternity ward does not allow a man to enter and see a laboring woman other than his delivering partner.*" (IDI-5, female health care worker)

Another health care provider added:

> "*It is unethical to invite a man in the labor room to be with his laboring partner while there are other laboring women in the same room which had no private doors*". (IDI-20, female health care worker)

## Shortage of medical supplies and demand for bribes

In Tanzania, maternal and child health services are offered free. These services include drugs and supplies. During the focus group discussions and interviews, it was noted that, when it happens there were a shortage of some of the services, pregnant women were required to purchase from their own pocket so that they can be attended by health care providers. Participants reported that men were asked to bring items required for delivery such as gloves, 'makintosh', and razor blade. Some of the dishonest health care providers even take it for granted to ask for bribes from men promising to supply their partners with all the items needed for delivery or give them priority in receiving the services. Participants also reported that if a pregnant woman was not accompanied by her partner, she might receive all the services free. This created obstacle to some men's involvement in maternal health care services for fear of being asked money for the services supposedly offered free. The implication of potential demand for money when seeking care at the health care facility was illustrated in one of the focus groups discussions:

> "*There is one thing that I noted here in our dispensary last month when I brought my in-law. It seems that health care providers need to be bribed for you to receive services. Especially when you appear to be smart they will ask you to provide some refreshment to them. This happened when I brought my patient here one of my relative told me that, the way you have dressed today health care providers will not let you go without giving them "soda". And it happened that they asked me to give money so that my patient can get a bed quickly since there was a long waiting list.*" (FGD-1, male participant)

Indirect cost was another health care system barrier that could influence men to accompany their partners or otherwise. It was noted that if men were unable to afford the items required for their partners to give birth, they would feel ashamed and would not want to accompany their partners to attend the health care facility for delivery. This view was shared by a female health care provider:

> "*Lack of working tools such as gloves leads some people to think that there is no need to have hospital delivering. This happens if they are asked to buy such essential tools.*" (IDI-11, female health care worker)

In one of the focus group discussions, a male participant said:

> "*If you accompany your wife to deliver at the health facility, they won't take care of her unless you give them extra money. They would ask you to buy syringes, razor blade and many other types of stuffs, which I even don't remember them. If you were not well prepared it's very shameful to be with her at the hospital*" (FGD-4, male participant)

## Discussion

This study explored community perspectives on barriers to men's involvement in maternity care, employing data-driven thematic analysis. Three themes prominently feature in the data including men's maternity care involvement indicators, benefits of men's involvement in maternity health care services and barriers to men's involvement in the maternity health care services. The first theme suggests avenues for men's involvement in the maternity health care and its measurability using two aspects of such involvement: attendance at the health care

facility during antenatal, delivery and postpartum care and household workload support during pregnancy. The second theme underscores the participants' support of the cruciality of men's attendance at the maternity health care service facilities with their partners by pointing out the benefits to be gained in terms of quality of maternity care to women. The third theme reveals an underlying systemic nature of barriers to men's involvement in the maternity health care services.

In the first theme, both men and women participants generally realize the importance of men's involvement in the maternity health care services; yet few men actually get involved. Most manifest measure of this realization is seen in both men's and women's perspectives which support men's attendance at the health care facilities, generally attributed to men's conventional gender role as providers for and protectors of their families. There is no doubt that accompanying a partner to ANC facility is more public; thus a man's reputation as a family protector may be at stake than workload support to a pregnant partner at home. All the same, very few men showed up with their partners at the maternity health care facilities beyond the necessary first appointment. Even much less men actually relieved their pregnant partners of household workload for reasons elaborated further, later.

The second theme data set points out benefits to be gained in terms of quality of maternity care to women. However, men's concerns about cultural beliefs and what their peers would say to them when getting involved in maternity care hampered the implementation of the professed beneficial move toward minimizing risks involved in maternity care at both home and the health care service facilities. Majority of male participants reported not understanding why they are called upon to attend at maternal health care services with their partners, while they never saw their parents did the same. Previous studies reported similar findings [16, 44–46]. The health sector is expected to be the source for maternal health information for men, disseminating appropriate information to men is important for their decisions on getting involved in maternity care [47].

Findings of this study suggest that barriers to male involvement in the maternity health care are systemic; starting from the family through the health care facilities, peer groups or neighbours to normative gender behaviour surrounding maternity and structural blind spots in the maternal health care policy in Tanzania. While at the family level men can afford not to get involved in the maternity health care of their partners for different reasons including lack of alternative care giver for small children, family financial constraints or dreading to be viewed as controlled/weak by fellow men when escorting a pregnant partner to the health care facility, no pregnant and/or labouring women can shed it off as they are biologically and socially responsible. Not only do they carry babies in their wombs, but also society entrusts women with care of infants. At the health care facilities, women endure all sorts of difficulties just to access maternity health care services, while male partners' discouragement seems justified even to women partners given circumstances such as distance to health care facility, care providers' attitude, delayed service, unfriendly consultation rooms, to mention just a few. Covert consequence is that, excluding men from caring for their partners lead to their feeling of uselessness, helplessness, anxiety and confusion, which may result to unhealthy couple relationships [24]. The intervention seeking to address reproductive health needs of both partners by engaging and educating men on care-seeking practices for themselves and their children may motivate men to be involved in maternity care issues. Health care providers have a big role to play in either promoting or discouraging men from participating in maternal and child health services [48]. Other studies also reported that harsh treatment of men by health care providers discouraged them from returning or participating in maternity care activities [45, 49, 50]. Some men felt uninvited when they managed to spare time to attend antenatal clinic [51] and harbored grudges toward partners for dragging them into such harsh maternity health care

environment. Health care facilities need to be approachable as when men feel that their presence is unacknowledged they are unlikely to repeat the visit or encourage their social group to do so. Therefore, health care providers need to be trained in interpersonal communication skills in order to change their attitudes towards men who accompany their partners.

Doubtlessly, couples must make difficult choices when confronted with challenges of splitting time between earning daily bread and companionship at the maternity health care facility; financial constraints; long waiting time at or tracking long distance to the health care facility. Specifically, men in the paid workforce often are not in a position to spend virtually the entire day participating in ANC services [5]. In such situations, women bear the brunt of the burden as they truly need the service. This finding is similar to other studies in Africa including Tanzania [33, 34, 45, 50]. While long distance to a health care facility is one of the most significant determinants in the decision on whether or not to seek modern health care services even when the person is in need [52], long waiting time at the health care facility strongly influences men's attendance at the maternity health care services [48]. This calls for the Ministry of Health, Community Development, Gender and the Elderly (MoHCDGE) to evaluate the success of Primary Health Services Development Program (PHSDP-MMAM) which addressed the delivery of health services within five kilometers to ensure fair, equitable and quality services to the community and increasing the number of health care providers in health care facilities so as to reach the WHO standard waiting time of less than one hour at a care centre. In so doing, it will ease such compounded burden of maternity health care services on the part of women.

Unethical behaviour among some health care providers who expect extra money beyond official fees before providing services deemed free in Tanzania, coupled with shortage of drugs and supplies in the government health care facilities discourage men escorting their partners to the clinic. This is more of a problem for those who cannot afford than those with means. Often than not, those with means can either use fast track services where queues are hardly there or tip their way into prompt service without wasting time. It is not uncommon for certain clients to be allowed to jump queues when others wait for too long to get the service. This finding concurs with those of previous studies done in Malawi and Tanzania that, when men escort their partners to the hospital they are told . . . to buy supplies like a basin, a razor blade and a plastic sheet, which discourage them [34, 45]. In Uganda for instance, some health care providers charged extra money beyond the official ANC fees to bridge their own financial gaps [5]. Thus health care providers need to be reminded to abide by their professional and ethical code of conduct and the government needs to maintain adequate supplies of basic needs for the ANC units as per the specific SDG number three (3).

At the community level and society in general, established gender norms foster a perspective that maternity is predominantly a women's affair, discouraging men's involvement in maternity health care services. Men's roles are mainly viewed as those of providers for or protectors of their families if there is a serious risk. Some household chores like cooking and infant care are also considered female responsibility. This finding concurs with previous studies in many African countries including Ghana, Kenya, Malawi, and Tanzania, which reported that men's role during pregnancy is that of provider; pregnancy support, infant care and cooking are particularly feminine roles and activities [13, 14, 32, 38, 44, 45, 49, 53, 54]. Men dread to be viewed as being overpowered/controlled by their partners (locally known as *bushoke*), bewitched or given love portion (locally known as *limbwata*). These views not only discourage men from involving in maternity care of their partners but also embarrass female partners; consequently forcing the latter to shoulder maternity care alone to save their male partners from humiliation as well as to maintain household harmony. Therefore, the existing policy for male involvement in maternal health should address the existing cultural practice and beliefs,

which create barriers for men's involvement in maternity care. Interventions should aim at creating awareness on the role of men in maternal health care, particularly in supporting their pregnant partners in taking control of their lives to inspire and hasten men's involvement in maternity health care services.

Tiny consultation rooms or lack of privacy in labor wards aside from being a health care facility issue inhibiting men's involvement in the maternity care of their partners is actually a policy issue spanning to societal level. It reveals the irony in the Tanzanian health care system agenda, which advocates men's involvement in maternal health care from pregnancy, delivery and postpartum periods as well as in child health monitoring [55], while the state of health care facilities remains the same. Labor wards have no birth cubicles, rendering laboring women to give birth in the open, witnessed by other laboring peers. Not only is this humiliating to individual women but also erodes any claim of professionalism within the health care system. Study participants have rightfully questioned, "how come public toilets have cubicles but not maternity wards?" This finding supports those of other studies in Malawi, Tanzania and South Africa [34, 38, 56]. It calls for the MoHCDGE through its health sector reforms, to seriously consider the issue of privacy in the delivery of maternal health care services to accommodate men's involvement agenda as well as the dignity of the women. It is long overdue for the Ministry to have birth cubicles in all maternity wards throughout the country.

Similarly, this study found HIV testing as part of the requirements of focused antenatal care package to be a barrier for both men and women to attend at ANC. It was noted that men felt obligated to attend ANC with their partners on the first ANC visit so that their partners may not be denied services when unaccompanied, even as some dreaded HIV testing. In subsequent visits, women either went alone or accompanied by their female relatives. Being a policy issue, it is worth pointing out that denying services to women who attend without their partners creates a barrier to women utilizing the services should their partners refuse to accompany them to the ANC visit. It may lead to late booking for ANC or no attendance at all, which may result to a negative pregnancy outcome. Other studies in Tanzania and other African countries had similar findings [34, 36, 38, 57, 58]. The good thing is that, women found ways of booking themselves for ANC, albeit challenges of their male partner reluctance. In order to reduce this barrier, disseminating proper information about the importance and benefits of men's involvement during ANC, beyond HIV testing as well as appropriate voluntary counseling and testing is crucial.

At this juncture, it is worth noting that strict demarcation between individual and systemic factors seems impossible when barriers to men's involvement in maternity health care are concerned. An individual male partner may not get involved in maternity health care because he doesn't understand, is not informed why he should be involved or is buying into gender norms discouraging his involvement. In any case, what starts with an individual can flare up to a group: a family, community or society. Awareness, information or knowledge may improve men's involvement in maternity health care. A previous study in Tanzania found that men who reported to have access to information regarding their involvement in maternity care were more likely to be involved compared to those who reported no access to information [47]. Thus health care providers in collaboration with community leaders need to create community awareness on men's involvement agenda and to combat the held negative traditional beliefs that discourage men's involvement in maternal health care. Similarly, some men who are not present at the ANC service may doubt accuracy of what their partners tell them concerning adequate nutrition, rest and birth preparedness; they may not appreciate the importance of giving the necessary support to their pregnant partners [27], or they may just not afford meeting those directives due to family social and economic situations, the latter which fall under systemic factors. It is also possible that women may not accurately communicate to

their partners the healthcare directives given at the clinic or consider them a luxury knowing their precarious family situations. Falnes et al., reported that some Tanzanian men see it as an insult to take instructions from women [27]. As much as it is true that, the directives may be easily implemented if a man discusses them directly with health care providers, in most cases clear understanding is certainly not the only barrier. Thus, it is important that, all factors be considered when planning intervention to improve men's involvement in maternity health care services. These include but not limited to, equipping male partners with right information and knowledge on the importance of providing adequate and effective support to pregnant partners, provision for affordable day care for small children and alleviating family poverty. Of course, lack of knowledge on maternal health issues impedes men on what they should do in order to support their partners, but most serious barriers linger around differential resource access that forces the women to sacrifice a lot in terms of nutrition, reducing workload and the necessary rest. As much as we agree with one scholar correctly arguing that, men expressed frustrations about not knowing what actions to take due to lack of knowledge on maternal issues [11], in our view for men to be more helpful, they certainly need material resources to implement such knowledge on maternity health care.

Equally important to note is the fact that some women may ignore inviting their male partners to attend at the health care facility for reasons best known to them including but not limited to, the legitimate partner not being responsible for the pregnancy or quietly contesting the hegemonic view of men controlling their lives or reproductive matter decisions. This study found pregnancy outside wedlock as one of the barriers to men's attendance at the health care facility. A further research is necessary to better understand and explain this seemingly aberrant phenomenon given that; most cultural communities in Tanzania generally enforce strict marital fidelity for women.

## Strengths and weaknesses of the study

The strength of this study is based on its data collection method and selection of study participants. In this study, focus group discussions and in-depth interviews were used as data collection methods. This method enabled the researchers to have a deeper understanding of men's involvement in the study area. Participants were drawn from different groups (men, women, health care providers and community leaders) and stratified by area both urban and rural residence, which captured diversity of study findings from different backgrounds. During the focus group discussions, participants may have been afraid to share some important information for fear of partner pressure. This challenge was addressed by using separate groups for male and female, and in addition, in-depth interviews were conducted to gain deeper understanding of community perspectives in relation to men's involvement. During transcription and translation of data, original meanings of data may have been missed. However, member checking, back translation and the use of moderator to proof read the transcripts was done in order to maintain the original meanings. The criteria used to include participants in the study might have led to social desirability and exclusion of single and unmarried parents assumed having limited experience with health care services as well as delivery and childbirth issues. Furthermore, the study did not consider the education background and employment status when selecting the participants. It would have been interesting to explore whether there are variations in perceptions of participants based on these characteristics. All authors had experience in working with community welfare and they were both involved in data collection and analysis; this could have created author's bias. However, the use of member checking technique increased the trustworthiness of the finding.

## Conclusion

Consideration of individual and systems factors is a key to understanding and addressing barriers to men's involvement in maternity health care. This study conceived men's involvement in maternity health care as attendance with partner at the health care facility and partner's workload relieve during pregnancy. The emerged interactive individual-systems factors influencing men's involvement in maternity health care include individual partners' personal qualities, their relationships as couples, family sizes and family economic conditions. Others are availability of accessible health care facility and its service environment in terms of schedule, staffing, supplies, professionalism, etc. Broader ones include culture-specific gendered expectation for maternity-related behaviour and not implementable policy provision on couple-friendly maternity health care services due to space limitation in consultation rooms and cubicle-less maternity wards.

These findings suggest that, conditions of deprivation in terms of tangible and non-tangible resources exacerbate not only men's non-involvement in their partners' maternity health care but also women's judgement and priorities regarding the best maternity health care practices throughout their reproductive life spans. Without undermining agency of individual partners, the implication of internalized deprivation is tending to alter self-worth, lowering expectations and normalizing deficiencies. This study noted that pregnant women shouldered normal or even more workload till delivery, attended antenatal care alone, and/or whenever necessary, explained justification for their partners' non-involvement in maternity care services. Women also gracefully endured harsh conditions of services at the maternity health care facilities, when their male partners often walked away.

The implications of these findings are three-fold: theoretical, practical/policy and further research. Theoretically, they have successfully fused interactive systems themes to explain barriers to men's involvement in maternity health care. For instance, patterns discerned from data suggest low attendance of male partners at the maternity health care facilities in general and impossibility of a daily wage earning men to accompany their partners to ANC visits in particular. Practically and policy-wise, it calls for interventions aimed at addressing structural aspects and conditions of deprivation along-side dissemination of couple-friendly knowledge and information on maternal health care. For further research, it recommends a follow up study to ascertain the breadth and depth of the aberrant finding on out of wedlock pregnancy within marital unions in Tanzania and their implication on men's involvement in maternity care. It also recommends evaluation of PHSDP-MMAM's success in increasing access to maternal health care services in general and men's involvement in maternity health care services in particular.

## Supporting information

**S1 Appendix. Focus group discussion guide.**
(PDF)

**S2 Appendix. Interview guide.**
(PDF)

**S3 Appendix. COREQ checklist.**
(DOC)

## Acknowledgments

We wish to acknowledge all those who agreed to participate in this study and the assistance provided by the staff of College of Health Sciences at the University of Dodoma.

## Author Contributions

**Conceptualization:** Nyasiro S. Gibore.

**Data curation:** Nyasiro S. Gibore.

**Formal analysis:** Nyasiro S. Gibore.

**Methodology:** Nyasiro S. Gibore, Theodora A. L. Bali.

**Supervision:** Theodora A. L. Bali.

**Validation:** Nyasiro S. Gibore, Theodora A. L. Bali.

**Visualization:** Nyasiro S. Gibore.

**Writing – original draft:** Nyasiro S. Gibore.

**Writing – review & editing:** Theodora A. L. Bali.

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
