## [Decision Letter · Decision Letter 0]

23 Dec 2019

PONE-D-19-26327

Community perspectives towards men’s involvement in maternity care: an exploration of potential barriers to men’s involvement in maternity care in a central Tanzanian community

PLOS ONE

Dear Dr., Gibore,

Thank you for submitting your manuscript to PLOS ONE. After careful consideration, we feel that it has merit but does not fully meet PLOS ONE’s publication criteria as it currently stands. Therefore, we invite you to submit a revised version of the manuscript that addresses the points raised during the review process.

We would appreciate receiving your revised manuscript by Feb 04 2020 11:59PM. To enhance the reproducibility of your results, we recommend that if applicable you deposit your laboratory protocols in protocols.io, where a protocol can be assigned its own identifier (DOI) such that it can be cited independently in the future. For instructions see: http://journals.plos.org/plosone/s/submission-guidelines#loc-laboratory-protocols

We look forward to receiving your revised manuscript.

Kind regards,

Calistus Wilunda, DrPH

Academic Editor

PLOS ONE

Additional Editor Comments:

The objective as stated in the abstract is slightly different from the one in the manuscript body. In the abstract, you mention “...to explore community perspectives on potential barriers toward men’s involvement…”, but in the main manuscript the objective is “…to provide opportunity for both men and women to share their experiences of male involvement…”.

Please explain more about data collection procedures in the manuscript. For instance, data collection tools are mentioned in the abstract but not in the main text. Include some of the information you have provided in the checklist you attached.

The checklist you provided is inadequate. Please include a 32-item COREQ checklist specifying the Item, Guide question, and the Page/section in your manuscript where this is reported. Refer to Table 1 of this paper: https://academic.oup.com/intqhc/article/19/6/349/1791966

Journal Requirements:

2. Please include a copy of the interview guide used in the study, in both the original language and English, as Supporting Information, or include a citation if it has been published previously.

Reviewers' comments:

Reviewer's Responses to Questions

**Comments to the Author**

1. Is the manuscript technically sound, and do the data support the conclusions?

Reviewer #1: Partly

Reviewer #2: Partly

2. Has the statistical analysis been performed appropriately and rigorously? 

Reviewer #1: N/A

Reviewer #2: N/A

3. Have the authors made all data underlying the findings in their manuscript fully available?

Reviewer #1: Yes

Reviewer #2: Yes

4. Is the manuscript presented in an intelligible fashion and written in standard English?

Reviewer #1: Yes

Reviewer #2: Yes

5. Review Comments to the Author

Reviewer #1: Recommendation: Major revision

Overarching comments

The authors work from a rich dataset and have reviewed a broad cross-section of relevant literature. However, in my opinion the study findings are not presented in a way that displays the analytical depth or precision that is expected from an academic article. While some substantial edits may be required to draw out the authors’ rich and nuanced findings, I believe that the authors’ methodology and conceptual focus have generated important findings that should be disseminated following revision.

Major revisions

The authors’ analysis and insights could be expressed more clearly in the manuscript. For example: “The effect of men’s involvement on health outcomes for women and children is directly linked to men’s knowledge, education, attitudes and behaviours” (Introduction, lines 49-51). What, specifically, do the authors mean here? The phrase “directly linked to” is not clear to me. Another example: the phrase “demonstrations of maleness” (Introduction, lines 56-57) is unclear. The authors engage with complex and nuanced concepts in the manuscript, and the language they use does not adequately express their points.

More broadly, in my opinion the manuscript does not engage in sufficient depth with the key issue of gender norms and gender power relations. The authors mention this issue, and to an extent situate their findings in the context of a patriarchal system. However, I believe that a deeper analysis is not clearly expressed in the current manuscript. For example, participants’ quotes illustrate a range of personal perspectives and community perspectives on male involvement, from an act of love to an act and practical/emotional support, to an act of jealousy, to a responsibility as head of the household, etc. However the authors do not address this diversity of rationales for male involvement, or the range of gender norms and gender power dynamics that inform these diverse perspectives. In a single paper it may not be feasible to engage deeply with the broad range of issues raised by the authors’ findings. I suggest that the authors clarify and narrow the aims and scope of the manuscript, in order to more deeply engage with their material.

Minor essential revisions

In my opinion the authors are inaccurate when they state that studies investigating barriers to men’s involvement in maternity care are limited (Introduction, line 74). There is a substantial and rapidly growing body of work in this area. Could the authors provide more detail on the specific gap in the literature that they have identified? The authors should also provide citations for some of the studies they mention that are focused on male involvement in PMTCT (Introduction, lines 72-73).

The study does not include the perspectives of single, unmarried, and first-time parents. The implications of this on study findings should be discussed as a limitation.

What are the “distinctive characteristics” of the study sites (Study area, line 112)? Additionally, qualitative sampling is typically designed to explore diversity, rather than to be representative, so the rationale for sampling diverse sites could be rephrased.

The themes ‘Women’s desire to have their partners in the antenatal clinic’, ‘Health care system barriers’, and ‘Location of the health care facilities’ would benefit from the addition of more illustrative quotes, to flesh out key findings and fully articulate the theme. For example, the single illustrative quote included in ‘Location of the health care facilities’ does not explain why this barrier impacts women’s male partners, but not women.

Discretionary, but recommended, revisions

The authors may wish to include a visual summary of their themes, to replace or complement the text summary in the first paragraph of the Results section.

The attributions for illustrative quotes (e.g. “FGD(2)-male participant Msalato_Dodoma municipal”) are distracting and could be formatted differently.

Reviewer #2: GENERAL COMMENTS

Thank you for giving me an opportunity to review this manuscript. The manuscript is relevant and focuses on an important public health issue. Overall, this manuscript suggests that the authors have undertaken a tremendous amount of work in the collection of data. The amount of interviews, and focus group discussions clearly demonstrate the amount of efforts the authors have invested on the collection of this data. However, there are some important issues which the authors need to address before the manuscript is accepted for publications.

Furthermore, the manuscript needs serious editing by a professional language editor. There are many editorial errors, unclear and or sensitive and inconsistently used words.

Title: The authors need to recast the title to make it short and clear. Delete unnecessary repetition of words in the title.

Abstract:

Line 26- need to be clear if they used content or thematic approach in data analysis. In the paper, they talk about thematic rather than content analysis.

Line 44 – chores instead of chore

INTRODUCTION

Major Comments

The concept male involvement should be comprehensively defined not just focusing on clinic attendance. Similarly, the discussion on male involvement should include psychological, medical, socio-cultural etc. The authors need to include empirical evidence on the benefits and potential harms of male involvement in maternal health care.

There is need to clearly indicate the knowledge gap and how the study advances existing knowledge.

Minor Comments

Line 49-51, recast this sentence. Similarly, line 56 -58.

Line 61, add word use or uptake after contraceptive.

Line 62, delete word maternal after increase

Line 66 – there is need also to highlight potential harms of male involvement in maternal health care.

Line 69-73 – recast this sentence.

Line 72 – the concept maternity care needs to be clearly defined.

MATERIALS AND METHODS

Major Comments

More information is needed on methods and data collection. For example, how were participants for the FGDs selected? What were the inclusion and exclusion criteria? Who conducted interviews and FGDs? Where were interviews and FGDs conducted? How were the number of interviews and FGDs determined - how saturation point was reached? How FGDs were organized? How quality was maintained during data collection? How was the Nvivo 9 software used in the data analysis.

More clarity is needed on the data analysis. The qualitative analysis part is rather simplistic and unclear. For example, who coded the data? How codes were determined? How was the coding done? Which themes emerged and how these themes were generated?

Minor Comments

Line 111 – 112 not clear what the authors mean.

Line 114 – 120 recast the paragraph, similarly, line 122 – 136.

RESULTS

Major Comments

This section is unnecessarily too long. There is need to reduce the number of quotes and only use them when necessary.

There is no table that show the themes and codes identified under each theme and makes it difficult to comment on how the codes are interpreted and how similar codes are categorized in to the same theme. However, some of the themes need to be merged to avoid repetition. For example, there are many repetition of what is discussed in the workload support and cultural barriers to male involvement.

While the authors indicate that they conducted 32 FGDs, throughout the results section all quotes only refer to FGD 1 and FGD 2. There is also confusion in many quotes. Sometimes FGD 1 indicate female participants and in other places male participants. Similarly, FGD 2 in some places they refer female participants while in other places male participants.

Minor Comments

Line 166 – 175 – need to summarize this paragraph

Line 315 – 317 – this theme needs to be expanded. It would also be good to get perceptions of men on this issue.

Line 370 – 380 – the quote is extremely long

Line 441 – 461- this is different from pregnancy outside marriage.

Line 493 – 513 – The description of this theme is not focused. Many issues are mixed together and thus difficult for the reader to understand.

Line 515 – this theme could better be named as lack of knowledge.

Line 528 – replace word nevertheless with an appropriate word.

Line 550 – make the theme short, it reads like a sentence rather than theme.

Line 582 – 586 – this theme needs to be expanded. Why women are able to attend health facilities despite the long distance? why not men? Are women in these communities physically stronger than men?

Line 607 –619- small consultation room is not discussed in the text. Only privacy in the labour room is discussed.

Line 621 – 640 – This theme is not clear. In Tanzania, maternal and child health services are nor charged. How then do shortage of medical supplies hinder men’s participation in maternal and child health services?

DISCUSSION

Overall, the discussion is weak and unnecessarily long. The discussion part is not different from the Finding. No new arguments that lead to conclusion. There is no critical analysis and reflection of the findings. Repeatedly, findings are compared with studies conducted in other countries. The authors need to discuss implications of the findings in Tanzania context.

Minor comments

Line 653 – 658 – These concepts should have been introduced in the results section. Don’t introduce new concepts in the discussion.

Pg 21- how are the two paragraphs different? Both seem to refer to cultural gender roles in the community.

Line 682 – 683 – Which study area are you referring to?

Line 683 – 686 – no need of repeating methods

Line 704 – 708 – This was not discussed in the results section. Don’t introduce new findings in the discussion.

Line 709 – 711 – The authors need to explain why women were able to attend despite long distance?

Line 711 – 715 – This strategy already ended in 2017 (2007 -2017). How can implementation then be accelerated?

Line 770 – 771 – No findings on the consultation rooms was reported in the results section.

Line 779 – 786 Need to refer to the correct name of the Ministry. This Ministry has changed since 4 years ago.

Line 788 – 791 This was not discussed in the results section. Don’t introduce new findings in the discussion.

STRENGTHS AND WEAKNESSES

How can findings of a qualitative study conducted in 16 villages be generalized for the entire country? The authors need to understand the purpose of conducting qualitative studies and why they differ from quantitative studies.

CONCLUSIONS AND RECOMMENDATIONS

There is no conclusion; the current conclusion is rather a summary of findings. The authors need to provide conclusions of the paper. Recommendations are generally weak. The authors should provide recommendations which are aligned to the findings of the study.

6. PLOS authors have the option to publish the peer review history of their article (what does this mean?). If published, this will include your full peer review and any attached files.

Reviewer #1: No

Reviewer #2: Yes: Stephen Maluka

---

## [Author Response · Author response to Decision Letter 0]

28 Feb 2020

Additional Editor Comments

Comments: The objective as stated in the abstract is slightly different from the one in the manuscript body 

Response: The objective in the main manuscript has been revised to match with research title reflected in the abstract See correction in page 7 line #173 to 175

Comment: Please explain more about data collection procedures in the manuscript. For instance, data collection tools are mentioned in the abstract but not in the main text. Include some of the information you have provided in the checklist you attached. 

Response: This section has been explained in details to include the important information that has been missing including; how were participants for the FGDs selected, inclusion and exclusion criteria, Who conducted interviews and FGDs, …. to mention the few. Inserted in pg 12 line #242 to 292

Comment:

The checklist you provided is inadequate. Please include a 32-item COREQ checklist specifying the Item, Guide question, and the Page/section in your manuscript where this is reported. 

Response: The checklist has been revised to include all 32-items and the page/section where it is reported has been indicated in the manuscript

Reviewer #1

Comment:The effect of men’s involvement on health outcomes for women and children is directly linked to men’s knowledge, education, attitudes and behaviours” (Introduction, lines 49-51). What, specifically, do the authors mean here? The phrase “directly linked to” is not clear to me. 

Response: The phrase “directly linked to” has been omitted and replaced with the phrase “associated with”See correction in page 4 line #93 to 94. Also two sentences has been added to the improve clarity of the paragraph.

(Inserted in pg 4 line #94 to 97)

Comments: The phrase “demonstrations of maleness” (Introduction, lines 56-57) is unclear. 

Response: This has been rephrased for better clarity as follows:

“In such situations where there is disregard of women’s views and deprivation of women’s access to household resources, women’s and children’s health status is affected.” See correction page 4 , line#102 to 105

Comment:The authors engage with complex and nuance concepts in the manuscript, and the language they use does not adequately express their points. I suggest that the authors clarify and narrow the aims and scope of the manuscript, in order to more deeply engage with their material.

Response: The manuscript has been revised to streamline arguments and improve clarity

Comment: In my opinion the authors are inaccurate when they state that studies investigating barriers to men’s involvement in maternity care are limited (Introduction, line 74). There is a substantial and rapidly growing body of work in this area. 

Response: Revisited the statement to mean… “However, studies investigating men’s involvement in maternity care and their barriers in Tanzania are limited” The word in Tanzania has been inserted in pg 6, line #144

Comment: Could the authors provide more detail on the specific gap in the literature that they have identified? 

Response: The section has been revised to clearly show the gap See rephrasing in Pg 7 line#168 to 172

Comment:The authors should also provide citations for some of the studies they mention that are focused on male involvement in PMTCT (Introduction, lines 72-73).

Response: The following citations has been inserted [10,25,28,34,51] Inserted in pg 6 line #143 to 144

Comment: The study does not include the perspectives of single, unmarried, and first-time parents. The implications of this on study findings should be discussed as a limitation.

Response: Single and unmarried parents were not part of our study population because we wanted to obtain information from couples who were living together. The first-time parents were included in the study as long as there were living together and had a child of age 2 or below years. The implication of excluding single and unmarried parents has been included in the limitations section (Inserted in page 52, line #1466 to 1468)

Comment:What are the “distinctive characteristics” of the study sites (Study area, line 112)? 

Response:This has been addressed, the paragraph has been added to elaborate those distinctive characteristics as seen in the manuscript (Inserted in line# 193 to 201 in pg 8)

Comment: Additionally, qualitative sampling is typically designed to explore diversity, rather than to be representative, so the rationale for sampling diverse sites could be rephrased.

Response: Revisited the statement to read “The districts were selected based to their diverse characteristics of each in relation to male involvement” See correction in page 8 line #191 to 193

Comment: The themes ‘Women’s desire to have their partners in the antenatal clinic’, ‘Health care system barriers’, and ‘Location of the health care facilities’ would benefit from the addition of more illustrative quotes, to flesh out key findings and fully articulate the theme. 

Response: The theme “Women’s desire to have their partners in the antenatal clinic” has been renamed as “Benefits of women having their partners at maternity health care services” (Inserted in page 20, line# 535). More illustrative quotes have been added as seen in the manuscript (Page 21, line# 559 to 568). The theme “Location of the health care facilities” has been renamed as “Distance to the health care facilities” (See page 32, line #904). More illustrative quotes have been added as seen in the manuscript (Page 33, line#956 to 960).

Comment: The authors may wish to include a visual summary of their themes, to replace or complement the text summary in the first paragraph of the Results section. 

Response: A visual summary of themes has been presented as per reviewer’s comment See Fig 1.

Comment:The attributions for illustrative quotes (e.g. “FGD(2)-male participant Msalato_Dodoma municipal”) are distracting and could be formatted differently. 

Response: The attributions for all illustrative quotes has been formatted sequentially from 1-32 as seen in the manuscript (See the results section pg 15 to 36, line #363 to 1054)

Reviewer #2

Comment: There are many editorial errors, unclear and or sensitive and inconsistently used words.

Response: The whole documents has been revised and editorial work has been done as seen in the manuscript

Comment: The authors need to recast the title to make it short and clear. Delete unnecessary repetition of words in the title.

Response: The title has been revised and it now reads as “Community perspectives: an exploration of potential barriers to men’s involvement in maternity care in a central Tanzanian community” (See corrected title page)

Comments: Line 26- need to be clear if they used content or thematic approach in data analysis. In the paper, they talk about thematic rather than content analysis.

Response:The study employed thematic approach during data analysis and the abstract has been revised accordingly 

(See corrected abstract)

Comment: Line 44 – chores instead of chore

Response: The whole section has been revised for better clarity. (See revision in page 2, line# 40 to 45)

Comment: The concept male involvement should be comprehensively defined not just focusing on clinic attendance. Similarly, the discussion on male involvement should include psychological, medical, socio-cultural etc. The authors need to include empirical evidence on the benefits and potential harms of male involvement in maternal health care Response: More explanation has been added to improve clarity as seen in the introduction section. (See revision in page 3 to 6, line# 66 to 136). For scope and manageability the study focused on two aspects namely; attendance in ANC and workload support during maternity period

Comment: There is need to clearly indicate the knowledge gap and how the study advances existing knowledge. Response: The section has been revised to clearly show the gap (Inserted in Pg 7 line#168 to 172)

Comment: Line 49-51, recast this sentence. Similarly, line 56 -58.

Response: The phrase “directly linked to” has been replaced with “associated with” and sentence had been recast. See correction in pg 4, line#93 to 97

Comment: Line 61, add word use or uptake after contraceptive. 

Response: The word ”use” has been inserted after the word “contraceptive” (Inserted in Pg 4 line#108)

Comment: Line 62, delete word maternal after increase 

Response: The word “maternal” has been removed. See correction in Page 5 line#109

Comment:Line 66 – there is need also to highlight potential harms of male involvement in maternal health care. Response: This has been addressed See addition in page 6, line# 127 to 136

Comment: Line 69-73 – recast this sentence. 

Response: The sentence has been recast (See correction page 6, line# 138 to 145)

Comment:Line 72 – the concept maternity care needs to be clearly defined.

Response: This has been defined as per reviewer’s comments (Inserted in Page 3, line#67 to 74)

Comment: More information is needed on methods and data collection. For example, how were participants for the FGDs selected? What were the inclusion and exclusion criteria? Who conducted interviews and FGDs? Where were interviews and FGDs conducted? How were the number of interviews and FGDs determined - how saturation point was reached? How FGDs were organized? How quality was maintained during data collection? How was the Nvivo 9 software used in the data analysis.

Response: The whole section has been revised to improve clarity of the manuscript (see revision in pg 10 to 12, line #242 to 292)

Comment: More clarity is needed on the data analysis. The qualitative analysis part is rather simplistic and unclear. For example, who coded the data? How codes were determined? How was the coding done? Which themes emerged and how these themes were generated?

Response: The whole section has been revised to improve clarity on data analysis as indicated in the manuscript (See correction in pg 12 to 13, line #297 to 331)

Comment:Line 111 – 112 not clear what the authors mean.

Response: The sentence has been rephrased to improve clarity (See correction in pg 8, line # 191 to 201)

Comment:Line 114 – 120 recast the paragraph, similarly, line 122 – 136.

Response:The paragraph has been revised, unclear statements have been omitted as seen in the manuscript (Omitted line #191 to 201 in page 8 to 9)

Comment: This section is unnecessarily too long. There is need to reduce the number of quotes and only use them when necessary.

Response: The number of quotes have been reduced and used only when necessary as seen in the manuscript (See the results section page 15 to 36, line #363 to 1054)

Comment: There is no table that shows the themes and codes identified under each theme and makes it difficult to comment on how the codes are interpreted and how similar codes are categorized in to the same theme. However, some of the themes need to be merged to avoid repetition. For example, there are many repetition of what is discussed in the workload support and cultural barriers to male involvement.

Response: Thematic figure showing codes, themes description and analytical themes has been prepared. Repetitive codes/themes have been merged (See fig 1.)

Comment: While the authors indicate that they conducted 32 FGDs, throughout the results section all quotes only refer to FGD 1 and FGD 2. There is also confusion in many quotes. Sometimes FGD 1 indicate female participants and in other places male participants. Similarly, FGD 2 in some places they refer female participants while in other places male participants.

Response:The illustrative quotes from FGDs have been sequentially presented from 1-32 as seen in the manuscript

(See the results section page 15 to 36, line #363 to 1054)

Comment: Line 166 – 175 – need to summarize this paragraph

Response: The whole section has been revised (See revision in page 15 to 16, line# 363 to 386)

Comment: Line 315 – 317 – this theme needs to be expanded. It would also be good to get perceptions of men on this issue.

Response: Revisited the theme to mean “Benefits of men's involvement in maternity health care services” The theme has expanded and perceptions of men has been included (See revision in page 20 to 21, line# 535 to 568)

Comment: Line 370 – 380 – the quote is extremely long 

Response:The illustrative quote has been shortened (See revision in page 23, line#623 to 630)

Comment:Line 441 – 461- this is different from pregnancy outside marriage.

Response: This section has been omitted (See page 27 to 28, line#775 to 795)

Comment:Line 493 – 513 – The description of this theme is not focused. Many issues are mixed together and thus difficult for the reader to understand.

Response: The theme has been renamed and it is now “Precarious family and economic situations” and only relevant illustrative quotes have been maintained and its description is now focused (See revision in page 28, line#833 to 856)

Comment: Line 515 – this theme could better be named as lack of knowledge. 

Response:The theme has been Revised to mean “Lack of knowledge and inadequate information” (See revision in pg 29, line# 857)

Comment:Line 528 – replace word nevertheless with an appropriate word.

Response: The word has been removed (See deletion page 30, line #886)

Comment:Line 550 – make the theme short, it reads like a sentence rather than theme 

Response: The theme name has been shorten to “Long waiting time” (See revision in page 31, line #908)

Comment:Line 582 – 586 – this theme needs to be expanded. Why women are able to attend health facilities despite the long distance? why not men? Are women in these communities physically stronger than men? 

Response: The theme has been expanded to clear the doubt (See correction in page 32 to 33, line#940 to 964)

Comment: Line 607 –619- small consultation room is not discussed in the text. Only privacy in the labour room is discussed. 

Response: The section has been revised for more clarity and relevant illustrative quote has been added (See correction in page 34, line#922 to 1011)

Comment: Line 621 – 640 – This theme is not clear. In Tanzania, maternal and child health services are not charged. How then do shortage of medical supplies hinder men’s participation in maternal and child health services?

Response: This has been addressed clearly in the manuscript (Seen revision in page 34 to 36, line#1013 to 1054

Comment: Overall the discussion is weak and unnecessarily long. The discussion part is not different from the Finding. No new arguments that lead to conclusion. There is no critical analysis and reflection of the findings. Repeatedly, findings are compared with studies conducted in other countries. The authors need to discuss implications of the findings in Tanzania context.

Response: The whole discussion section has been revised (See the discussion section)

Comment: Line 653 – 658 – These concepts should have been introduced in the results section. Don’t introduce new concepts in the discussion. 

Response: The concepts are in the results section (See the results section page 23 to 24, line#648 to 683)

Comment:Pg 21- how are the two paragraphs different? Both seem to refer to cultural gender roles in the community.

Response: The whole section has been revised including these paragraphs (See the discussion section)

Comment:Line 682 – 683 – Which study area are you referring to? 

Response: It was refereeing to central part of Tanzania. However this section has been revised (See revision in page 8, line# 189 to 201)

Comment: Line 683 – 686 – no need of repeating methods 

Response: This section has been revised (See the discussion section)

Comment:Line 704 – 708 – This was not discussed in the results section. Don’t introduce new findings in the discussion.

Response: This section has also been revised (See the revision in the discussion section)

Comment: Line 709 – 711 – The authors need to explain why women were able to attend despite long distance?

Response: The statement has been qualified in the discussion section (See the discussion section)

Comment: Line 711 – 715 – This strategy already ended in 2017 (2007 -2017). How can implementation then be accelerated? 

Response: The recommendation has been recast (See the revision in the discussion section page 39, line#1143)

Comment:Line 770 – 771 – No findings on the consultation rooms was reported in the results section. 

Response: The section has been revised and the relevant illustrative quote has been added (See revision in page 34, line#922 to 1011)

Comment: Line 779 – 786 Need to refer to the correct name of the Ministry. This Ministry has changed since 4 years ago.

Response: The ministry name has been corrected as “Ministry of Health, Community Development, Gender and the Elderly (MoHCDGE)” Seen correction in page 39, line#1143

Comment: Line 788 – 791 This was not discussed in the results section. Don’t introduce new findings in the discussion. Response: The section has been revised (See the discussion section)

Comment: How can findings of a qualitative study conducted in 16 villages be generalized for the entire country? The authors need to understand the purpose of conducting qualitative studies and why they differ from quantitative studies.

Response: The strengths and weaknesses section has been revised and relevant information has been added (See revision in page 52, line#1453 to 1475)

Comment: There is no conclusion; the current conclusion is rather a summary of findings. The authors need to provide conclusions of the paper. Recommendations are generally weak. The authors should provide recommendations which are aligned to the findings of the study.

Response: The conclusion and recommendation section has been revised, reflecting the theoretical and practical implication of the findings (See revision in page 53 to 54, line#1479 to 1514)

---

## [Decision Letter · Decision Letter 1]

6 Apr 2020

PONE-D-19-26327R1

Community perspectives: an exploration of potential barriers to men’s involvement in maternity care in a central Tanzanian community

PLOS ONE

Dear Dr., Gibore,

Thank you for submitting your manuscript to PLOS ONE. Peer review for this manuscript is now complete but I notice that you sometimes use the present tense in the results section (for example line 423) . Can you please revise this section accordingly. You can also use this opportunity to proofread the  entire manuscript  and correct any grammatical errors. For instance, you talk of "Men's attendance to maternity health services". I think it is correct to say "Men's attendance of maternity healthcare services..."

We would appreciate receiving your revised manuscript by May 21 2020 11:59PM. To enhance the reproducibility of your results, we recommend that if applicable you deposit your laboratory protocols in protocols.io, where a protocol can be assigned its own identifier (DOI) such that it can be cited independently in the future. For instructions see: http://journals.plos.org/plosone/s/submission-guidelines#loc-laboratory-protocols

We look forward to receiving your revised manuscript.

Kind regards,

Calistus Wilunda, DrPH

Academic Editor

PLOS ONE

Reviewers' comments:

Reviewer's Responses to Questions

**Comments to the Author**

1. If the authors have adequately addressed your comments raised in a previous round of review and you feel that this manuscript is now acceptable for publication, you may indicate that here to bypass the “Comments to the Author” section, enter your conflict of interest statement in the “Confidential to Editor” section, and submit your "Accept" recommendation.

Reviewer #2: All comments have been addressed

2. Is the manuscript technically sound, and do the data support the conclusions?

Reviewer #2: Yes

3. Has the statistical analysis been performed appropriately and rigorously? 

Reviewer #2: N/A

4. Have the authors made all data underlying the findings in their manuscript fully available?

Reviewer #2: Yes

5. Is the manuscript presented in an intelligible fashion and written in standard English?

Reviewer #2: Yes

6. Review Comments to the Author

Reviewer #2: The manuscript has improved significantly. I am happy that the authors have addressed all important comments I raised. I have not additional comments.

7. PLOS authors have the option to publish the peer review history of their article (what does this mean?). If published, this will include your full peer review and any attached files.

Reviewer #2: Yes: Stephen Maluka

---

## [Author Response · Author response to Decision Letter 1]

24 Apr 2020

Editor’s Comments

Comments:

I notice that you sometimes use the present tense in the results section (for example line 423). Can you please revise this section accordingly?

Response:

The section has been revised and is now in the past tense as seen in the manuscript 

Comment:

You can also use this opportunity to proofread the entire manuscript and correct any grammatical errors. For instance, you talk of "Men's attendance to maternity health services". I think it is correct to say "Men's attendance of maternity healthcare services..."

Response:

Editorial work has been done in the entire document as seen in the manuscript. However, we would prefer to use the word “at” instead of “of”. We think it is correct to say “Men’s attendance at maternity health care services” instead of “Men’s attendance of maternity health care services”. The correct grammar for attendance is “attendance at” and can’t get any other option

---

## [Editor Report · Decision Letter 2]

27 Apr 2020

Community perspectives: an exploration of potential barriers to men’s involvement in maternity care in a central Tanzanian community

PONE-D-19-26327R2

Dear Dr. Gibore,

We are pleased to inform you that your manuscript has been judged scientifically suitable for publication and will be formally accepted for publication once it complies with all outstanding technical requirements.

With kind regards,

Calistus Wilunda, DrPH

Academic Editor

PLOS ONE
---

## [Editor Report · Acceptance letter]

6 May 2020

PONE-D-19-26327R2 

Community perspectives: an exploration of potential barriers to men’s involvement in maternity care in a central Tanzanian community 

Dear Dr. Gibore:

I am pleased to inform you that your manuscript has been deemed suitable for publication in PLOS ONE. Congratulations! Your manuscript is now with our production department. 

With kind regards,

on behalf of

Dr. Calistus Wilunda 

Academic Editor

PLOS ONE